# Short Term Results of Early Treatment of Developmental Dysplasia of the Hip or Luxation with Pavlik Harness in Human Position

**DOI:** 10.3390/medicina58020206

**Published:** 2022-01-28

**Authors:** Manuel Gahleitner, Rainer Hochgatterer, Gerhard Großbötzl, Lorenz Pisecky, Matthias Klotz, Tobias Gotterbarm, Günter Hipmair

**Affiliations:** Department for Orthopedics and Traumatology, Kepler University Hospital GmbH, Johannes Kepler University Linz, 4040 Linz, Austria; Rainer.hochgatterer@kepleruniklinikum.at (R.H.); Gerhard.grossboetzl@kepleruniklinikum.at (G.G.); Lorenz.pisecky@kepleruniklinikum.at (L.P.); Matthias.klotz@kepleruniklinikum.at (M.K.); Tobias.gotterbarm@kepleruniklinikum.at (T.G.); Guenter.hipmair@kepleruniklinikum.at (G.H.)

**Keywords:** DDH, Pavlik harness, congenital dysplasia/luxation/human position

## Abstract

*Background and Objectives**:* This study shows a sufficient treatment with the Pavlik harness for all patients through all phases of developmental dysplasia of the hip (DDH) if there is a strict regime. *Materials and Methods*: There was an ultrasound measurement stage of IIc or worse (D, IIIa/b, IVa/b) in 159 out of 7372 newborns between 1995 and 2006 (2.15%). This is an indication for treatment with the Pavlik harness. Overall, 203 dysplastic hips were treated initially with our regime. After detection, we started the application of the Pavlik harness immediately in the ‘human position’. There were appointments every 10–14 days to check the setting combined with ultrasound controls. The treatment stopped if a mature, well-developed picture of both hips was seen when compared to Graf type Ia/b. Afterwards, an X-ray control was carried out at about one year of age. *Results*: 159 newborns with 203 dislocated hips were treated. The distribution following Graf’s classification was as follows: 150 type IIc (73.9%), 18 type D (8.9%), 31 type IIIa/b (15, 3%) and 4 type IV (1.9%). To summarize, there were 150 (73.9%) type IIc hips at risk of developing a dislocation but also 53 hips (26.1%) which were already dislocated at the moment of birth. There was a loss to follow-up in three patients (1.8%), and the therapy had to be changed in six cases. There was no degradation in our study population during therapy. *Conclusion:* The treatment with the Pavlik harness of DDH at every stage in newborns was possible and showed good results in 189 hips.

## 1. Introduction

Due to the introduction of ultrasound examination of newborn hips by Prof. Graf in 1980 [1,2], a screening method of developmental dysplasia of the hip (DDH) was created which is irreplaceable in Austria today. It was Pavlik, in 1957 [3], who treated the first newborn with developmental dysplasia of the hip with the Pavlik harness. There are a few publications [4,5,6] with alternative treatment regimens concerned with the time that such therapies begin. The treatment usually starts in the first week of a newborn’s life. The possible complication of iatrogenic femoral head necrosis was stated with the Pavlik harness in the “Lorenz position” up to 90° abduction with up to 15% [7,8], but only with up to 5% in the human position with 110° flexion and 45° abduction [9]. Due to the early start of the treatment, the possibility of manual repositions [10], as well as the postulated high growth potential of the orbital cartilage of the acetabulum area [11], could be used for healing. The hypothesis of this ongoing group study was that, based on current evidence, developmental dysplasia of the hip in early newborns could be treated with the Pavlik harness.

## 2. Materials and Methods

All newborns meeting essential criteria for treatment were included, meaning there was no control group or randomization trials. There is a mandatory hip sonography for every newborn child in Austria since 1991. It does not matter if the hips are stable or if there is any suspected pathology at the first clinical exam after birth. Due to this, 7372 newborns were screened from 1995 to 2006. For the sonographs, a GE Logiq 3 with 10-megahertz linear transducer GE 10LB (GE Healthcare GmbH, 42655 Solingen, Germany) and an obligatory splint Type SonoFix (Gebrüder Hirschbeck GmbH, 8843 St. Peter am Kammersberg, Austria) were used.

In the period mentioned above, there were 203 pathological hips (1.38%) requiring treatment using Graf’s classification for staging. For this project, all patients with developmental dysplasia of the hip due to neurological reasons were excluded.

A total of 159 (2.16%) newborns were affected, which included 150 Type IIc (73.9%), 18 Type D (8.9%), 31 Type IIIa/IIIb (15.3%) and finally 4 cases of Type IV. If we talk about Type D, we are talking about an already dislocated hip. Type IIc is mentioned as a hip at risk. So, if therapy does not get started, we expect a developing dislocation. At this point, it is important to mention that no distinction was made between IIc stable hips and IIc unstable hips (Table 1).

There was an affection of 102 right hips (50.5%) and 101 left hips (49.5%). In 51 cases (32.0%), both hips were affected. The gender breakdown showed 131 female (82.4%) and 28 male (17.6%) newborns with a pathological sonography.

If, in our performed hip sonography, there was a pathological result of Type IIc or worse, therapy with a Pavlik harness was started with a maximum flexion of 100 degrees and a maximum abduction of 45 degrees. This is called the human position. Therapy was started as soon as possible after birth. The difference between the human position (a) and Lorenz position (b) is shown in Figure 1. We used a Pavlik harness (Heindl, Linz, Austria). Checks were made in accordance with the defined scheme. The first check-up was done before discharge from our hospital, or within the first two weeks of starting the therapy. At these first check-ups, great importance was attached to the fit of the Pavlik harness, the skin appearance and the compliance of the parents. This was followed by an ultrasound check and a clinical check every 2–3 weeks. Parents were advised not to change the Pavlik position at home. In the event of the fitting becoming loose, the Pavlik harnesses were marked to ensure a good position. In the case of unsolvable problems, the parents could come to the outpatient clinic at any time.

The end of therapy was determined using the sonographic morphologically mature Type I hip according to the Graf classification.

At the beginning of the run, i.e., at the age of about one year, an anterior-posterior X-ray control was performed, among other things, to exclude an AVN.

## 3. Results

Therapy with a Pavlik harness was started on 159 newborns between 1995 and 2006. In at least one case, there was a change of our therapy after one week from a Pavlik harness to a Fettweis spica cast. The reason was that the birth weight was much too low (<1500 gr) and this made the smallest Pavlik harness too large.

Therapy was stopped, or rather changed, in another eight cases due to different reasons. Three cases were lost because the parents no longer appeared for inspection, and no reason was given. All in all, 150 newborns (94.34%) with 189 dysplastic hips reached the defined aim. Table 2 shows the number of therapy dropouts by year and frequency.

Type IIc and D had an average treatment duration of 50 days between 1995 and 2006. Type IIIa/b and IV showed an average treatment duration of approximately 65 days, meaning two weeks longer than lighter forms of congenital hip dysplasia, such as IIc. The average duration of therapy was 53 days to reach the defined goal (Alpha > 60°) of the whole study group. This is mentioned in Table 3.

In this study, there was not a single case of femoral head necrosis at the age of one year or the beginning of the run. Occasional mild skin irritation in the area of the hip bend could be treated with local therapy. In just one case, the therapy had to be changed to a Tuebingen hip flexion splint because of a femoral ulcer.

## 4. Discussion

Developmental dysplasia of the hip, or dislocation, is reported with an incidence rate of 2 to 4 percent for Central Europe [12,13]. The pure dislocation rate (Type D or worse) is reported at 0.5 to 1 percent. The results of this study correspond to the results of other mentioned publications. Hip dysplasia occurred in 2.16% (159 of 7372 newborns) and hip dislocation in 0.72% (53 of 7372 newborns) of newborns. In addition, the ratio of female to male, at almost 4:1, corresponds to the known figures from previous studies [14,15,16]. As postulated by some authors, manual reduction is only possible in the first 10–14 days, provided there is no intra-acetabular mechanical obstacle. In 1994, Tschauner described the growth potential of the endochondral ossification with a corresponding maturation curve [11]. There is a high form differentiation in the first 6 weeks of life, which flattens as early as week 12 and reaches a plateau phase between weeks 12 and 16. The possibility of ossification of the acetabular cartilage occurs especially in the first 6–12 weeks of life. We see these two facts as the basis for carrying out our early start of therapy.

All dislocated hips can be non-invasively and manually repositioned by using this method, and an average treatment duration of 53 days across all types of dysplasia or dislocation is an acceptable period for parents, and we are therefore able to maintain unrestricted treatment. The complete maturation of highly dislocated hips such as type III or IV also showed a nice treatment result in a period of approximately 10 weeks. Some studies show a longer therapy duration [17].

At first, the parents were sceptical of the Pavlik harness but the alternatives, plaster pants such as the Fettweis spica cast, were always rejected.

The second very popular conservative treatment method for DDH is the Tübingen hip flexion splint. The difference to the treatment with the Pavlik harness is mainly in the area of application over clothing and in the somewhat easier handling. The results of a study by Ran et al. indicate that patients with severe forms of DDH (Graf grade IV) and bilateral hip involvement treated with the Tübingen hip flexion splint had a higher failure rate than patients treated with the Pavlik harness [18]. In contrast to this, Kubo et al. stated in 2017 that in terms of results there is no difference between the Pavlik harness, Tübingen hip flexion splint and Fettweiss spica cast [17]. X.Lyu et al. also stated that no significant difference in success rate was found in the dysplasia group, nor for infants who were treated prior to 90 days of age with either the Pavlik harness or Tübingen hip flexion splint. Furthermore, the effect of time on hip development following reduction from either form of treatment was compared by X.Lyu et al. and showed no significant difference in the overall rate of recovery [19].

There was no case of deterioration at the ultrasound check-ups. In conclusion, we consider the early treatment of primary congenital hip dysplasia or dislocation with the Pavlik harness as a very safe therapy with few side effects. Our results compare well with other studies [19,20,21,22,23,24,25,26].

The hip-sonography screening method is essential for early treatment and should be done during the first week of life. If therapy is started on time, there is no complication such as a femoral head necrosis or an invasive hip reduction.

We postulate that screening as early as possible (first week of life) with subsequent treatment of all dysplasia stages with a Pavlik harness represents a safe form of therapy with few side effects.

The average duration of treatment for hip dysplasia or dislocation Type III or IV according to Graf’s classification can be specified at approximately 10 weeks in our patient population. All newborns treated later than the first week of life (external assignments) showed a longer treatment period. In this group, where treatment was later than the first week, we sometimes had to perform an invasive reduction. For this reason, and as confirmed by our research data, we affirm a screening within the first week of life, and the start of treatment with a Pavlik harness for congenital hip dysplasia where necessary.

## 5. Conclusions

These results confirm that unstable hips such as Graf type III and IV can also be treated with a Pavlik harness and show a good result in the one-year follow-up. Furthermore, there was no avascular femur head necrosis in our population after the one-year follow-up.

## Figures and Tables

**Figure 1 medicina-58-00206-f001:**
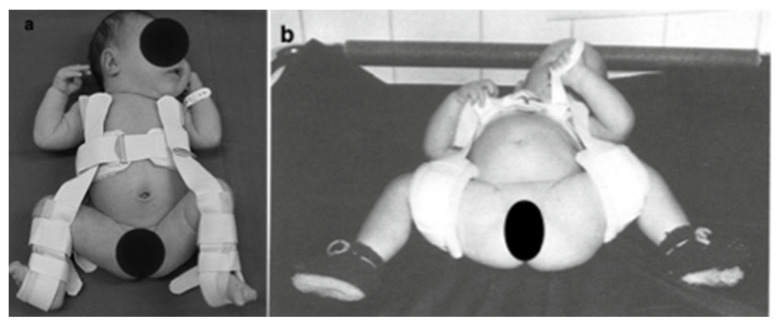
(**a**) is current from our setting. This is how the Pavlik harness is applied in human position. (**b**) Instead of the Pavlik harness, a Daimler bandage is used. The difference between the human position and the Lorenz position is clearly visible in the abduction up to more than 90° [12].

**Table 1 medicina-58-00206-t001:** Shows the side, gender and stage distribution of the population in treatment years 1995 to 2006.

Year	New Born	I and IIa	Therapy			IIc, D, III, IV	Right	Left	Both	Pavlik Harness in %	Dis- Located in %	IIC	D	III	IV
				♀	♂										
1995	589	574	15	12	3	22	11	11	7	2547	1868	9	4	9	0
1996	715	703	12	7	5	16	6	10	4	1678	1119	10	1	3	2
1997	680	670	10	7	3	15	7	8	5	1471	1103	11	2	2	0
1998	529	511	18	17	1	23	11	12	6	3403	2174	23	0	0	0
1999	487	467	20	19	1	28	14	14	6	4107	2875	24	1	3	0
2000	672	654	18	14	4	15	8	7	4	2679	1116	13	0	2	0
2001	640	627	13	9	4	14	7	7	1	2031	1094	11	1	1	1
2002	628	612	16	13	3	21	10	11	6	2548	1672	17	2	2	0
2003	543	535	8	6	2	9	5	4	1	1473	0.829	9	0	0	0
2004	611	597	14	13	1	21	11	10	7	2291	1718	10	2	8	1
2005	666	661	5	5	0	6	4	2	1	0.751	0.450	3	2	1	0
2006	612	602	10	9	1	13	7	6	3	1634	1062	10	3	0	0
			159	131	28	203	101	102	51			150	18	31	4

**Table 2 medicina-58-00206-t002:** Number of therapy dropouts by year and frequency.

Year	Patients	Reason for Changing Therapy	Initial Diagnosis
1995	1	Ulcus femoral—Change to Mittelmayer Spreizhose	IIc right
	1	Change to Tuebingen hip flexion splint at IIa ipsilateral	IIIa ipsilateral
	1	Change to Tuebingen hip flexion splint after 16 weeks due to compliance troubles	IIIa left
1996	1	Changed therapy to a Fettweis spica cast because of the much too low birth weight	IV ipsilateral
1998	1	Pavlik harness was removed from mother at Type IIc ipsilateral	IIc ipsilateral
1999	1	Mother no longer appeared for inspection after 4 weeks of therapy at Type IIa	IIc ipsilateral
2001	1	Change to Tuebingen hip flexion splint after 3 weeks of therapy at Type IIa ipsilateral	IIc ipsilateral
2002	1	Parents no longer appeared after the first check-up	IIc right
2004	1	Change to Tuebingen hip flexion splint after 5 weeks due to compliance troubles	IIc right

**Table 3 medicina-58-00206-t003:** Average duration of therapy in days to reach the defined goal.

Year	Average Duration of Therapy	Minimum	Maximum
1995	67.4 days	28 days	182 days
1996	65.5 days	42 days	147 days
1997	70.8 days	21 days	105 days
1998	49.4 days	28 days	84 days
1999	52.3 days	28 days	140 days
2000	45.5 days	28 days	63 days
2001	49 days	28 days	63 days
2002	46.1 days	21 days	85 days
2003	46 days	28 days	56 days
2004	63.5 days	42 days	112 days
2005	40.6 days	49 days	70 days
2006	40.5 days	21 days	66 days
	53.05 days		

## Data Availability

The collected data are properly stored in the study archive of the Department of Orthopaedics and Traumatology at Kepler University Hospital and protected from misuse by third parties.

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
