# Peer review of "Short Term Results of Early Treatment of Developmental Dysplasia of the Hip or Luxation with Pavlik Harness in Human Position"

_medicina, 2022, doi:10.3390/medicina58020206_

Round 1

Reviewer 1 Report

I read this interesting article titled “Short term results of early treatment of congenital hip dysplasia or luxation with Pavlik harness in human position”. The authors present the results of their DDH treatment strategy application of Pavlik harness after the first screening visit in "human position". Even I prefer the usage of Frejka pillow for IIc stable hips in early phases rather than Pavlik harness, I accept their strategy to simplify the possibilities of therapy options.

 Few issues were outlined as the following:  

  • It would be instructional to provide a scheme or photo of a patient with a Pavlik Harness in a human position and distinguish it from a Lorenz-position
  • Please mention the name of the company, producing Pavlik Harness for the baby above 1500g (unavailable in my conditions) in the text.
  • I would prefer Developmental Dysplasia of the hip (DDH) in the title and in the manuscript rather than Congenital Hip Dysplasia (DDH) 

Author Response

Dear reviewer 1.

Thanks for the detailed comments. I hope I can give the appropriate answers and clear up any ambiguities. I have of course also changed this in the publication and treated all stated points.

1.) I tried to find any photograph of a newborn child with Pavlik harness in Lorenz position to distinguish it from human position. I just was able to find a suitable picture in a Daimler-Bandage but I think the difference is visible. 

2.) Our Pavlik harness are manufactured by Heindl company in Linz, Upper Austria. The bandages are sewn by hand, but meet all the requirements of the European Medical Devices Act and are approved. They are available in sizes XS to L and thus fit even for very small newborns.

3.) I changed throughout the manuscript to the more appropriate term.

I hope you like my update. 

Kind regards

Dr. Gahleitner

Reviewer 2 Report

 the material & methods need to be specific: why was the US performed; were there any clinical signs to point out DDH; when was the harness applied; for how much time was it continued; what were the instructions given to parents on the harness; how was exact 45degrees abduction maintained on the harness: what signs were used to detect the AVN

No statistical methods have been used

discussion is not comprehensive. pl do not repeat information given in introduction. it must include comparison with other conservative methods; details of complications & reference to age & treatment matched studies.

Author Response

Dear reviewer 2.

Thanks for the detailed comments. I hope I can give the appropriate answers and clear up any ambiguities. I have of course also changed this in the publication and treated all stated points.

1.) There is a mandatory hip sonography for every newborn since 1991. So it doesn't matter if there is no pathological sign like Ortolani or Barlow. Every newborn gets two hip sonographies - one in the first week of their lives and a second one between week 6 and 8. This is also prescribed in the so called "Mutter-Kind-Pass". So if there is a pathological alpha-angle and a development dysplasia of the hip at the first hip sonography, we started our therapy - best, of course, as soon as possible after birth. Therapy continued until an alpha angle of at least 60° was achieved, representing a mature hip according to Graf. 
The instructions for the parents were clearly given. The Pavlik harness are not to be changed. If a brackets comes loose, we mark the braces so that the parents can independently close them again in the correct position at home. 

At the beginning of the run, i.e. at the age of about one year, an anteriorposterior X-ray control was performed - among other things also to exclude an AVN.

We tried to adjust the 45° abduction as good as possible over the length of the individual braces, in this aspect the controls also take place very closely (every two weeks)

In coordination with the statistical institute of the Johannes Kepler University Linz (KKS), purely descriptive statistics were used for this project.

I have adjusted the discussion and hope that it now corresponds to your comments. 

I would be happy about a positive feedback.

Kind Regards

Dr. Gahleitner

Round 2

Reviewer 2 Report

May be accepted in present form

This manuscript is a resubmission of an earlier submission. The following is a list of the peer review reports and author responses from that submission.